# PERMUTATION INVARIANT GRAPH-TO-SEQUENCE MODEL FOR TEMPLATE-FREE RETROSYNTHESIS AND REACTION PREDICTION

## ABSTRACT

Synthesis planning and reaction outcome prediction are two fundamental problems in computer-aided organic chemistry for which a variety of data-driven approaches have emerged. Natural language approaches that model each problem as a SMILES-to-SMILES translation lead to a simple end-to-end formulation, reduce the need for data preprocessing, and enable the use of well-optimized machine translation model architectures. However, SMILES representations are not an efficient representation for capturing information about molecular structures, as evidenced by the success of SMILES augmentation to boost empirical performance. Here, we describe a novel Graph2SMILES model that combines the power of Transformer models for text generation with the permutation invariance of molecular graph encoders that mitigates the need for input data augmentation. As an end-to-end architecture, Graph2SMILES can be used as a drop-in replacement for the Transformer in any task involving molecule(s)-to-molecule(s) transformations. In our encoder, a~~n attention-augmented~~ directed message passing neural network (D-MPNN) captures local chemical environments, and the global attention encoder allows for long-range and intermolecular interactions, enhanced by graph-aware positional embedding. Graph2SMILES improves the top-1 accuracy of the Transformer baselines by $1.7\%$ and $1.9\%$ for reaction outcome prediction on USPTO_480k and USPTO_STEREO datasets respectively, and by $9.8\%$ for one-step retrosynthesis on the USPTO_50k dataset.

## 1 INTRODUCTION

Retrosynthetic analysis (Corey, 1988; Corey & Cheng, 1989) and its reverse problem, reaction outcome prediction (Corey & Wipke, 1969), are two fundamental problems in computer-aided organic synthesis. The former tries to propose possible reaction precursors given the desirable product, whereas the latter aims to predict the major products given reactants. Historically, they were tackled using rule-based expert systems such as LHASA (Corey et al., 1972). Recent developments in machine learning have led to a number of new template-based, graph edit-based, and translation-based methods, for which we give a detailed review in Section 3.1. For both tasks, translation-based approaches have grown popular, possibly because the end-to-end formulation is procedurally simple. Since most organic molecules can be represented as SMILES strings (Neglur et al., 2005), retrosynthesis can be cast as a translation from product SMILES to reactant SMILES (Liu et al., 2017), and so can reaction outcome prediction (Nam & Kim, 2016). Modeling these tasks as machine translation problems enables the use of neural architectures that are well-studied and well-optimized in the field of Natural Language Processing (NLP). Several of the best performing models across multiple benchmark datasets (Tetko et al., 2020; Irwin et al., 2021; Sun et al., 2020; Seo et al., 2021; Wang et al., 2021b) have used the Transformer architecture (Vaswani et al., 2017) as the backbone on SMILES representations, showing the effectiveness of translation-based formulation.

However, SMILES representations do not provide bijective mappings to molecular structure. As a result, data augmentation with chemically-equivalent SMILES (Bjerrum, 2017) has become a common practice to improve empirical performance. Augmenting the training data with just 1 equivalent reaction SMILES by permuting both the inputs and the outputs can already produce noticeable improvements of $0.8\%$ to $4.3\%$ (Schwaller et al., 2019; Seo et al., 2021). Incorporating 9 (Wang et al.,

2021b), or up to 100 (Tetko et al., 2020) equivalent SMILES can provide additional gains. While these efforts demonstrate the effectiveness of SMILES augmentation, this introduces a non-trivial choice as to how much augmentation should be done. The performance has not saturated even with 100 augmented SMILES (Tetko et al., 2020), but this may significantly complicate the model pipeline, especially during test time which typically necessitates de-duplication, ensembling, and heuristic-based scoring (Wang et al., 2021b; Tetko et al., 2020).

In this paper, we propose a novel graph-to-sequence architecture called Graph2SMILES to solve the tasks of retrosynthesis and reaction prediction. We first design a sequential graph encoder with an attention-augmented directed message passing neural network (D-MPNN) based on Yang et al. (2019) and optionally augmented by attention-based message updates, followed by a Transformer-based global attention encoder with graph-aware positional embedding. We then pair the graph encoder with a Transformer decoder to transform molecular graph inputs into SMILES outputs, without using sequence representations of input SMILES at all. As such, we guarantee the permutation invariance of Graph2SMILES to the input, eliminating the need for input-side augmentation altogether. Our main contributions can be summarized as follows:

1. We propose a Graph2SMILES architecture with two encoder components modeling local and global atomic interactions respectively, to address forward prediction and one-step retrosynthesis as graph-to-sequence tasks.

2. We design a graph-aware positional embedding to further enhance performance. It is easily generalizable to graphs containing two or more molecules, while not requiring any pre-training or joint training with auxiliary tasks.

3. We demonstrate the adequacy of graph representations alone by showing that Graph2SMILES outperforms Transformer baselines on predictive chemistry tasks without needing any input-side SMILES augmentation.

Our Graph2SMILES architecture achieves state-of-the-art top-1 accuracy on common benchmarks among methods that do not make use of reaction templates, atom mapping, pretraining, or data augmentation strategies. We emphasize that Graph2SMILES is a backbone architecture, and hence a drop-in replacement for the Transformer model. As such, Graph2SMILES can be plugged into any method for molecular transformation tasks that uses the Transformer model, while retraining the benefits from techniques or features orthogonal to the architecture itself.

## 2 METHODS

### 2.1 GRAPH AND SEQUENCE REPRESENTATIONS OF MOLECULES

There are multiple ways of representing molecular structures, such as by molecular fingerprints (Rogers & Hahn, 2010), by SMILES strings (Weininger, 1988), or as molecular graphs with atoms as nodes and bonds as edges. We represent the input molecule(s) as graphs, and the output molecule(s) as SMILES strings, thus modeling both reaction prediction and retrosynthesis as graph-to-sequence transformations.

Formally, let $\mathcal{G}_{in}$ denotes the molecular graph input, which can contain multiple subgraphs for different molecules, with a total of $N$ atoms. Following the convention in Somnath et al. (2020), we describe $\mathcal{G}_{in} = (\mathcal{V}, \mathcal{E})$ with atoms $\mathcal{V}$ and bonds $\mathcal{E}$. Each atom $u \in \mathcal{V}$ has a feature vector $\boldsymbol{x}_u \in \mathbb{R}^a$, and each directed bond $(u, v) \in \mathcal{E}$ from atom $u$ to $v$ has its feature vector as $\boldsymbol{x}_{uv} \in \mathbb{R}^b$. The details of the atom and bond features used can be found in Appendix A. We build the input molecular graphs from their SMILES strings with RDKit (Landrum, 2016). Note that all feature vectors are invariant to the order of atoms and bonds, as well as to how the original SMILES strings are written. We represent the output as a sequence of SMILES tokens $\mathcal{S}_{out} = \{s_1, s_2, \ldots, s_n\}$, where the tokens $\{s_i\}$ are obtained from the canonical SMILES using the regex tokenizer in Schwaller et al. (2019).

### 2.2 GRAPH2SMILES

The Graph2SMILES model is a variant of the encoder-decoder model (Cho et al., 2014) commonly used for machine translation. Figure 1 displays the architecture of Graph2SMILES with the permutation invariant graph encoding process shown within the blue dashed box. We replace the

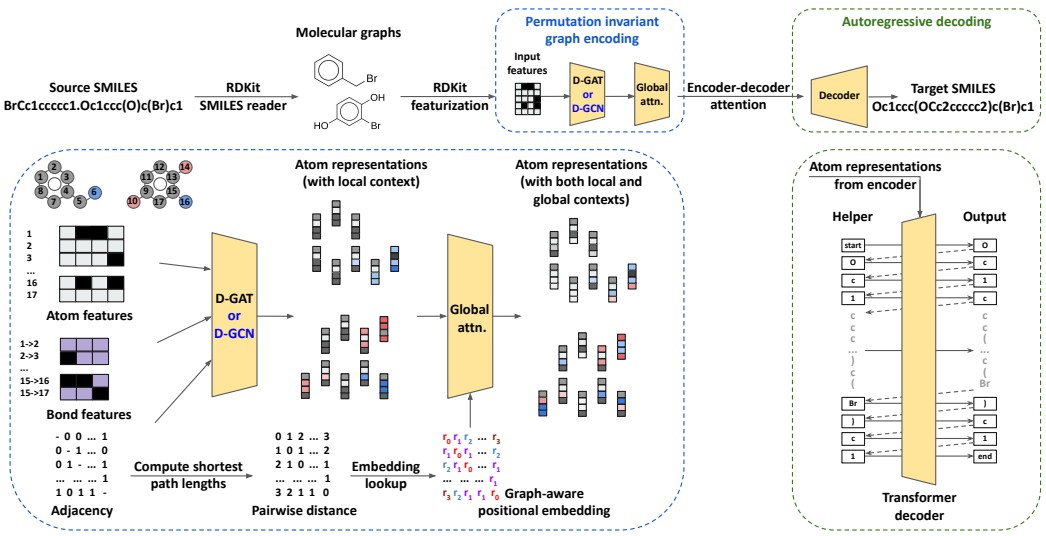

Figure 1: Model architecture for Graph2SMILES. Top: the overall flowchart. Bottom left: details of permutation invariant graph encoding. Bottom right: details of autoregressive decoding.

encoder part of the standard Transformer model (Vaswani et al., 2017) used in Molecular Transformer (Schwaller et al., 2019) with a ~~novel attention-augmented~~ directed message passing encoder, followed by a global attention encoder with carefully designed graph-aware positional embedding. Each module has its intuitive function: the D-MPNN captures the local chemical context, the global attention encoder allows for global-level information exchange, and the graph-aware positional embedding enables the attention encoder to make use of topological information more explicitly. The permutation invariant encoding process eliminates the need for SMILES augmentation for the input side altogether, simplifying data preprocessing and potentially saving training time.

### 2.2.1 ATTENTION AUGMENTED DIRECTED MESSAGE PASSING ENCODER

The first module of the graph encoder is a D-MPNN (Yang et al., 2019) with Gated Recurrent Units (GRUs) (Cho et al., 2014) used for message updates (Jin et al., 2018; Somnath et al., 2020). Unlike atom-oriented message updates in edge-aware MPNNs (Hu et al., 2020; Yan et al., 2020; Mao et al., 2021; Wang et al., 2021a), updates in D-MPNN are oriented towards directed bonds to prevent totters, or messages being passed back-and-forth between neighbors (Mahé et al., 2004; Yang et al., 2019). We optionally augment the D-MPNN with attention-based message updates inspired from Graph Attention Network (Veličković et al., 2018; Brody et al., 2021). We term this variant as Directed Graph Attention Network (D-GAT) and refer to the original D-MPNN variant used in Somnath et al. (2020) as Directed Graph Convolutional Network (D-GCN), while keeping both D-GAT and D-GCN as possible design choices for the D-MPNN.

For D-GAT, at each message passing step $t$, the message $\boldsymbol{m}_{uv}^{t+1}$ associated with each directed bond $(u, v) \in \mathcal{E}$ is updated using

$$\boldsymbol{s}_{uv} = \text{AttnSum}\left(\boldsymbol{x}_u, \boldsymbol{x}_{uv}, \left\{\boldsymbol{m}_{wu}^t\right\}_{w \in N(u) \backslash v}\right) \tag{1}$$

$$\boldsymbol{z}_{uv} = \sigma\left(\boldsymbol{W}_z\left[\boldsymbol{x}_u; \boldsymbol{x}_{uv}; \boldsymbol{s}_{uv}\right] + \boldsymbol{b}_z\right) \tag{2}$$

$$\boldsymbol{r}_{uv} = \sigma\left(\boldsymbol{W}_r\left[\boldsymbol{x}_u; \boldsymbol{x}_{uv}; \boldsymbol{s}_{uv}\right] + \boldsymbol{b}_r\right) \tag{3}$$

$$\tilde{\boldsymbol{m}}_{uv} = \tanh\left(\boldsymbol{W}\left[\boldsymbol{x}_u; \boldsymbol{x}_{uv}\right] + \boldsymbol{U}\boldsymbol{r}_{uv} + \boldsymbol{b}\right) \tag{4}$$

$$\boldsymbol{m}_{uv}^{t+1} = (1 - \boldsymbol{z}_{uv}) \odot \boldsymbol{s}_{uv} + \boldsymbol{z}_{uv} \odot \tilde{\boldsymbol{m}}_{uv} \tag{5}$$

where $\boldsymbol{W}_z, \boldsymbol{W}_r, \boldsymbol{W}, \boldsymbol{U}$ and $\boldsymbol{b}_z, \boldsymbol{b}_r, \boldsymbol{b}$ are the learnable weights and biases respectively. $\sigma$ is the sigmoid function, ";" indicates concatenation, and $\odot$ is the element-wise product. AttnSum is the attention-based message aggregation defined as

$$e_{wu} = \boldsymbol{a}^T \text{LeakyReLU}\left(\boldsymbol{W}_{qk}\left[\boldsymbol{x}_u; \boldsymbol{x}_{uv}; \boldsymbol{m}_{wu}^t\right] + \boldsymbol{b}_{qk}\right) \qquad (6)$$

$$a_{wu} = \frac{\exp\left(e_{wu}\right)}{\sum_{w \in N(u)\backslash v} \exp\left(e_{wu}\right)} \qquad (7)$$

$$\text{AttnSum}\left(\boldsymbol{x}_u, \boldsymbol{x}_{uv}, \left\{\boldsymbol{m}_{wu}^t\right\}_{w \in N(u)\backslash v}\right) = \sum_{w \in N(u)\backslash v} a_{wu}\left(\boldsymbol{W}_v \boldsymbol{m}_{wu}^t + \boldsymbol{b}_v\right) \qquad (8)$$

where $\boldsymbol{W}_{qk}, \boldsymbol{b}_{qk}, \boldsymbol{a}$ are the learnable parameters for the attention scores, and $\boldsymbol{W}_v, \boldsymbol{b}_v$ are the parameters for the value vectors. We have omitted a self-loop in message aggregation, as it did not have any noticeable effect on performance from our experiments. In Eqn (3) we simplify the reset gate to be shared for all incoming edges, in contrast to Somnath et al. (2020) where a separate reset gate is defined for each edge.

Lastly, after $T$ iterations, we obtain the atom representations $\boldsymbol{h}_u$ with similar attention-based aggregation (with different parameters) over the bond messages coming into each atom $u$, followed by a single output layer with weight $\boldsymbol{W}_o$ and GELU activation (Hendrycks & Gimpel, 2016).

$$\boldsymbol{m}_u = \text{AttnSum}'\left(\boldsymbol{x}_u, \left\{\boldsymbol{m}_{wu}^{(T)}\right\}_{w \in N(u)}\right) \qquad (9)$$

$$\boldsymbol{h}_u = \text{GELU}\left(\boldsymbol{W}_o\left[\boldsymbol{x}_u; \boldsymbol{m}_u\right]\right) \qquad (10)$$

We use multi-headed attention in our formulation similar to Brody et al. (2021).

### 2.2.2 GLOBAL ATTENTION ENCODER WITH GRAPH-AWARE POSITIONAL EMBEDDING

To capture global interactions, the atom representations coming out of the D-MPNN are fed into a global attention encoder, which is a variant of the Transformer encoder. We incorporate graph-aware positional embedding, adapted from the relative positional embedding used in Transformer-XL (Dai et al., 2019b) as follows. Firstly, in the standard Transformer either with sinusoidal encoding (Vaswani et al., 2017; Schwaller et al., 2019) or learnable (Devlin et al., 2019) absolute positional embedding, the attention score between atoms $u$ and $v$ can be decomposed as

$$e_{u,v}^{abs} = \boldsymbol{h}_u^T \tilde{\boldsymbol{W}}_q^T \tilde{\boldsymbol{W}}_k \boldsymbol{h}_v + \boldsymbol{h}_u^T \tilde{\boldsymbol{W}}_q^T \tilde{\boldsymbol{W}}_k \boldsymbol{p}_v + \boldsymbol{p}_u^T \tilde{\boldsymbol{W}}_q^T \tilde{\boldsymbol{W}}_k \boldsymbol{h}_v + \boldsymbol{p}_u^T \tilde{\boldsymbol{W}}_q^T \tilde{\boldsymbol{W}}_k \boldsymbol{p}_v \qquad (11)$$

where $\tilde{\boldsymbol{W}}_q, \tilde{\boldsymbol{W}}_k$ are weights for the keys and queries, and $\boldsymbol{p}_u, \boldsymbol{p}_v$ are the absolute positional encoding or embedding corresponding to atoms $u$ and $v$. Similar to Transformer-XL, we reparameterize the four terms. Instead of using sequence-based relative positional embedding $\boldsymbol{r}_{u-v}$, we use a learnable embedding term $\boldsymbol{r}_{u,v}$ that is dependent on the shortest path length between $u$ and $v$

$$e_{u,v}^{rel} = \left(\boldsymbol{h}_u^T \tilde{\boldsymbol{W}}_q^T + \boldsymbol{c}^T\right) \tilde{\boldsymbol{W}}_k \boldsymbol{h}_v + \left(\boldsymbol{h}_u^T \tilde{\boldsymbol{W}}_q^T + \boldsymbol{d}^T\right) \tilde{\boldsymbol{W}}_{k,R} \boldsymbol{r}_{u,v}$$

$$= \left(\boldsymbol{h}_u^T \tilde{\boldsymbol{W}}_q^T + \boldsymbol{c}^T\right) \tilde{\boldsymbol{W}}_k \boldsymbol{h}_v + \left(\boldsymbol{h}_u^T \tilde{\boldsymbol{W}}_q^T + \boldsymbol{d}^T\right) \tilde{\boldsymbol{r}}_{u,v} \qquad (12)$$

The trainable biases are renamed as $\boldsymbol{c}$ and $\boldsymbol{d}$ to avoid confusion and shared across all layers. Intuitively, the two terms in Eqn (12) model the interactions between inputs ($\boldsymbol{h}_u$ and $\boldsymbol{h}_v$), and between input and relative graph position ($\boldsymbol{h}_u$ and $\tilde{\boldsymbol{r}}_{u,v}$) respectively. Unlike Transformer-XL, we forgo the inductive bias built into the sinusoidal encoding, merging $\tilde{\boldsymbol{W}}_{k,R} \boldsymbol{r}_{u,v}$ into a single learnable $\tilde{\boldsymbol{r}}_{u,v}$. This makes the relative positional embedding easily generalizable to atoms not within the same molecule, which is particular useful for reaction outcome prediction and, more generally, tasks with more than two input molecules. We also bucket the distances similar to Raffel et al. (2020) such that

$$\mathcal{B}_{u,v} = \begin{cases} \text{distance}(u, v), & \text{if distance}(u, v) < 8 \\ 8, & \text{if } 8 \leq \text{distance}(u, v) < 15 \\ 9, & \text{if } 15 \leq \text{distance}(u, v) \text{ and } u, v \text{ are in the same molecule} \\ 10, & \text{if } u, v \text{ are not in the same molecule} \end{cases}$$

$\mathcal{B}_{u,v}$ is then used to look up $\tilde{\boldsymbol{r}}_{u,v}$ in the trainable positional embedding matrix. The rest of the global attention encoder mostly follows a standard Transformer with multi-headed self-attention (Vaswani et al., 2017), layer normalization (Ba et al., 2016), and position-wise feed forward layers, except no positional information is added to the value vectors within each Transformer layer.

### 2.2.3 Sequence decoder

We use a Transformer-based autoregressive decoder to decode from the atom representations after the global attention encoder. Each output token is generated by attending to all atoms with encoder-decoder attention (Bahdanau et al., 2015; Vaswani et al., 2017), while also attending to all tokens that have already been generated. Following Seo et al. (2021), we set max relative positions to 4 for the decoder, thereby enabling the usage of sequence-based relative positional embedding used in Shaw et al. (2018) and implemented by OpenNMT (Klein et al., 2017).

### 2.3 Model training

The Graph2SMILES model is then trained to maximize the conditional likelihood

$$p\left(\mathcal{S}_{\text{out}}|\mathcal{G}_{\text{in}}\right) = p\left(s_1, s_2, \ldots, s_n | \mathcal{G}_{\text{in}}\right) = \prod_{i=1}^{n} p_\theta\left(s_i | s_{1:i-1}, \mathcal{G}_{\text{in}}\right) \tag{13}$$

## 3 Related work

### 3.1 Reaction outcome prediction and one-step retrosynthesis

One approach to computer-aided reaction prediction and retrosynthesis is to make use of chemical reaction rules based on subgraph pattern matching that are formalized as reaction templates, as in expert systems such as LHASA (Corey et al., 1972) and SYNTHIA (Szymkuć et al., 2016). More recent efforts have used neural networks to model the two tasks as template classification (Segler & Waller, 2017; Baylon et al., 2019; Dai et al., 2019a; Chen & Jung, 2021), or template ranking based on molecular similarity (Coley et al., 2017). These template-based approaches select the top ranked templates, which can then be applied to transform the input molecules into the outputs.

For template-based methods, there is an inevitable tradeoff between template generality and specificity. Further, these methods cannot generalize to unseen templates. As a remediation for such intrinsic limitations, a number of template-free approaches have emerged over the recent years, which can be categorized into graph edit-based and translation-based. The first category models reaction prediction or retrosynthesis as graph transformations (Jin et al., 2017; Coley et al., 2019; Do et al., 2019; Bradshaw et al., 2019; Sacha et al., 2021; Qian et al., 2020). Variants of graph edit methods include electron flow prediction (Bi et al., 2021) and semi template-based methods where reaction centers are first identified, followed by a graph or sequence recovery stage (Shi et al., 2020; Yan et al., 2020; Somnath et al., 2020; Wang et al., 2021b). Translation-based formulations, on the other hand, approach the problems as SMILES-to-SMILES translation, typically with sequence models such as Recurrent Neural Networks (Nam & Kim, 2016; Schwaller et al., 2018; Liu et al., 2017) or the Transformer (Schwaller et al., 2019; Lin et al., 2020; Lee et al., 2019; Duan et al., 2020; Tetko et al., 2020). Variants of these approaches design additional stages such as pretraining and reranking (Irwin et al., 2021; Zhu et al., 2021; Zheng et al., 2020; Sun et al., 2020), or use information about graph topology to enhance performance (Yoo et al., 2020; Seo et al., 2021; Mao et al., 2021).

Our approach is similar to GET (Mao et al., 2021) which also solves one-step retrosynthesis with graph-enhanced encoders and sequence decoders. Unlike GET which concatenates the SMILES sequence embeddings and learned atom representations, thereby not guaranteeing permutation invariance, we do not use the sequence representation at all in our encoder. Yet Graph2SMILES yields significant improvement over GET on USPTO_50k, demonstrating the power of our graph encoder and the adequacy of graph representations alone.

### 3.2 Adapting the Transformer encoder for molecular representation

The idea of modeling graphs using the Transformer architecture is not new. In the domain of molecular representation, the Molecular Attention Transformer (Maziarka et al., 2020) and GeoT (Kwak et al., 2021) inject atomic distance information into when computing attention scores. However, the computation of such distance information itself requires sampling of 3D conformers, thereby introducing an additional source of variations and breaking order invariance. An alternative is using the lengths of shortest paths between atoms. GRAT (Yoo et al., 2020) uses these lengths to parameterize the optional scale and bias for the attention score, whereas PAGTN (Chen et al., 2019)

treats them as path features in its additive attention. Our use of pairwise shortest path lengths between atoms for our Transformer-based global attention encoder are inspired by GRAT and PAGTN. Contrary to their usage of these lengths as additional features, we explicitly separate the effect of graph topology using graph-aware relative positional embedding, considering the success of such specially designed embedding in graph representation (Ying et al., 2021) and other domains (Shaw et al., 2018; Dai et al., 2019b; Wang et al., 2019; Guo et al., 2020). Our formulation of relative positional embedding builds on top of its counterpart in Transformer-XL (Dai et al., 2019b), which we found to be empirically superior than using single learnable bias as in T5 (Raffel et al., 2020) and Graphormer (Ying et al., 2021).

### 3.3 COMBINATION OF GRAPH NEURAL NETWORKS AND TRANSFORMER

Combining graph encoder and Transformer encoder in a sequential manner has been explored in GET (Mao et al., 2021) and NERF (Bi et al., 2021), as well as in Wang et al. (2021a) and GROVER (Rong et al., 2020) albeit for different molecular learning tasks. Graph2SMILES does not use the sequence representation as an input as in GET, or require the output molecular graph in the encoder as in NERF. Also, none of these related works retain the explicit information about graph topology before passing the atom representations into the attention encoder like we do, which we show to be important in the ablation study in Section 4.5. Similarly, the graph-to-sequence formulation itself has been used in NLP for conditional text generation tasks such as SQL-to-text (Xu et al., 2018a;b) and AMR-to-text (Cai & Lam, 2020). Our encoder is different from these prior studies; most notably, the graph-aware positional embedding is designed to easily generalize to more than two disconnected graphs, which is typical for reaction outcome prediction.

## 4 EXPERIMENTS

### 4.1 DATASETS

We evaluate model performance by top-n test accuracies on four USPTO datasets derived from reaction data originally curated by Lowe (2012). The details of these datasets are summarized in Appendix C. For reaction outcome prediction, we evaluate on the USPTO_480k_mixed and USPTO_STEREO_mixed datasets following Schwaller et al. (2019). The suffix _mixed indicates that the reactants and reagents have not been separated based on which species contribute heavy atoms to the product. While USPTO_480k has been preprocessed by Jin et al. (2017), USPTO_STEREO was filtered to a lesser extent, retaining stereochemical information and reactions forming or breaking aromatic bonds. For one-step retrosynthesis, we evaluate on the USPTO_full and USPTO_50k datasets without reaction type, both of which have been used as benchmarks for retrosynthesis. We count a prediction as correct only if it matches the ground truth output SMILES exactly, including all stereochemistry but excluding atom mapping, after canonicalization by RDKit.

### 4.2 IMPLEMENTATION DETAILS

For D-GAT and D-GCN, we change the hidden size to 256 from 300 used in GraphRetro (Somnath et al., 2020) to be more consistent with the global attention encoder. The number of message updating steps is set to 4. For the global attention encoder, following Molecular Transformer and GTA, we fix the embedding and hidden sizes $d_{model}$ to 256, the filter size for Transformer to 2048, the number of attention heads to 8, and the numbers of layers for both the attention encoder and the Transformer decoder to 6. We train our model using Adam optimizer (Kingma & Ba, 2015) with Noam learning rate scheduler (Vaswani et al., 2017). Similar to Schwaller et al. (2019), we group reactions with similar number of SMILES tokens together, batch by the maximal number of token count, and scale the D-MPNN outputs by $\sqrt{d_{model}}$ before feeding into the attention encoder. The details of the hyperparameters used for different datasets are summarized in Appendix D. We save the model checkpoints every 5000 steps, select the best checkpoints based on the top-1 accuracy on the validation sets, and report the performance on the held-out test sets. Beam search is used to generate the output SMILES during inference with a beam size of 30. We filter out any SMILES that cannot be parsed by RDKit and keep the remaining as our final list of proposed candidates for evaluation.

### 4.3 RESULTS ON REACTION OUTCOME PREDICTION

Table 1 summarizes the results of Graph2SMILES and other existing works on reaction outcome prediction. We only include methods that perform evaluations on the USPTO_480k_mixed dataset in the table, and provide a brief comparison of other methods that only evaluate on the less challenging USPTO_480k_separated data in Appendix B. All excluded methods show inferior performance to the Molecular Transformer (MT), with the exception of NERF (Bi et al., 2021) which has a 0.3 point improvement in top-1 accuracy.

Table 1: Results for reaction outcome prediction on USPTO_480k_mixed and USPTO_STEREO_mixed. Best results for complete columns are highlighted in **bold**.

| Methods | Top-$n$ accuracy (%) | | | |
|---|---|---|---|---|
| | 1 | 3 | 5 | 10 |
| **USPTO_480k_mixed** | | | | |
| MEGAN (Sacha et al., 2021) | 86.3 | 92.4 | 94.0 | 95.4 |
| Molecular Transformer (Schwaller et al., 2019) | 88.6 | 93.5 | 94.2 | 94.9 |
| Graph2SMILES (D-GCN) (*ours*) | 90.3 | 94.0 | 94.6 | 95.2 |
| Graph2SMILES (D-GAT) (*ours*) | 90.3 | 94.0 | 94.8 | 95.3 |
| Augmented Transformer (Tetko et al., 2020) | 90.6 | - | **96.1** | - |
| Chemformer (Irwin et al., 2021) | **91.3** | - | 93.7 | 94.0 |
| **USPTO_STEREO_mixed** | | | | |
| Molecular Transformer (Schwaller et al., 2019) | 76.2 | 84.3 | 85.8 | - |
| Graph2SMILES (D-GAT) (*ours*) | 78.1 | 84.5 | 85.7 | 86.7 |
| Graph2SMILES (D-GCN) (*ours*) | **78.1** | **84.6** | **85.8** | 86.8 |

For the USPTO_480k_mixed dataset, Graph2SMILES improves upon the MT baseline with 1.7, 0.5 and 0.4 point increases in top-1, 3 and 10 accuracies respectively. Similarly, for the USPTO_STEREO_mixed dataset, Graph2SMILES improves the top-1 accuracy of the MT baseline by 1.9 points, with minor improvement for top-3 accuracy. While there is still a gap between Graph2SMILES and Augmented Transformer or Chemformer, our approach does not use test-time data augmentation and ensembling as in Augmented Transformer (Tetko et al., 2020), nor have we performed pretraining as in Chemformer (Irwin et al., 2021) whose models have up to 10 times as many parameters as Graph2SMILES. Ensembling and pretraining are potential directions for improving Graph2SMILES as they are still compatible with our backbone replacement for the Transformer. For reaction outcome prediction, there is a small advantage of using D-GAT over D-GCN, with improvements of up to 0.2 points on top-n accuracy.

### 4.4 RESULTS ON ONE-STEP RETROSYNTHESIS

We compare the results of one-step retrosynthesis on USPTO_full of Graph2SMILES with all existing methods that report results on this dataset, to the best of our knowledge. As can be seen from Table 2, Graph2SMILES achieves higher top-1 accuracy than all methods except GTA (Seo et al., 2021), while not using any templates, atom mapping, or output-side data augmentation. These additional features or techniques, which have been demonstrated to improve the performance of Transformer variants (Appendix E), are orthogonal to the graph-to-sequence architecture itself, and can therefore potentially improve Graph2SMILES as well. For example, atom mapping can be used to enhance Graph2SMILES with graph-truncated cross-attention as in GTA.

The much smaller USPTO_50k dataset has been benchmarked more extensively. We compare the results in Table 3, marking only the usage of the same set of features and techniques used as in Table 2 for succinctness[1]. The first group of rows shows that across methods that do not use reaction

---

[1] Graph2SMILES can also benefit from other techniques such as latent variable modeling (Chen et al., 2020), which we demonstrate in Appendix E. From Table 10, using latent classes N = 2 can already boost the top-10 accuracies of the D-GCN variant from 72.9 to 79.5, and of D-GAT from 73.9 to 77.7.

Table 2: Retrosynthesis results on USPTO_full. *Templ.*: reaction templates used; *Map.*: atom-mapping required; *Aug.*: output data augmentation used. Best results are highlighted in **bold**.

| Methods | Top-$n$ accuracy (%) | | Features / techniques used | | |
|---|---|---|---|---|---|
| | 1 | 10 | *Templ.* | *Map.* | *Aug.* |
| RetroSim (Coley et al., 2017) | 32.8 | 56.1 | ✓ | ✓ | ✗ |
| MEGAN (Sacha et al., 2021) | 33.6 | 63.9 | ✗ | ✓ | ✗ |
| NeuralSym (Segler & Waller, 2017) | 35.8 | 60.8 | ✓ | ✓ | ✗ |
| GLN (Dai et al., 2019a) | 39.3 | 63.7 | ✓ | ✓ | ✗ |
| Transformer baseline (Zhu et al., 2021) | 42.9 | 66.8 | ✗ | ✗ | ✗ |
| RetroPrime (Wang et al., 2021b) | 44.1 | 68.5 | ✗ | ✓ | ✓ |
| Aug. Transformer (Tetko et al., 2020) | 44.4 | **73.3** | ✗ | ✗ | ✓ |
| DMP fusion (Zhu et al., 2021) | 45.0 | 67.9 | ✗ | ✗ | ✗ |
| Graph2SMILES (D-GAT) (*ours*) | 45.7 | 62.9 | ✗ | ✗ | ✗ |
| Graph2SMILES (D-GCN) (*ours*) | 45.7 | 63.4 | ✗ | ✗ | ✗ |
| GTA (Seo et al., 2021) | **46.6** | 70.4 | ✗ | ✓ | ✓ |

templates, atom mapping, or output SMILES augmentation, Graph2SMILES achieves the best top-1 accuracy, improving the Transformer baseline (Lin et al., 2020) by 9.8 points from 43.1 to 52.9. From our experiments, we observe that it is possible to boost top-n accuracies for $n > 1$ at the expense of sacrificing the top-1 accuracy (even slightly), creating some room for tradeoff. To avoid over-tuning and giving overly optimistic results, however, we only report the test results for models with the highest top-1 accuracy during validation. Unlike in reaction outcome prediction, using D-GAT does not have a clear advantage over D-GCN for retrosynthesis, yielding only improvement on top-5 and 10 accuracies for USPTO_50k as in Table 3.

Table 3: Retrosynthesis results on USPTO_50k without reaction type. *Templ.*: reaction templates used; *Map.*: atom-mapping required; *Aug.*: output data augmentation used. Best results for each group of rows are highlighted in **bold**.

| Methods | Top-$n$ accuracy (%) | | | | Features / techniques used | | |
|---|---|---|---|---|---|---|---|
| | 1 | 3 | 5 | 10 | *Templ.* | *Map.* | *Aug.* |
| AutoSynRoute (Lin et al., 2020) | 43.1 | 64.6 | 71.8 | **78.7** | ✗ | ✗ | ✗ |
| GET (Mao et al., 2021) | 44.9 | 58.8 | 62.4 | 65.9 | ✗ | ✗ | ✗ |
| DMP fusion (Zhu et al., 2021) | 46.1 | 65.2 | 70.4 | 74.3 | ✗ | ✗ | ✗ |
| Tied Transformer (Kim et al., 2021) | 47.1 | **67.2** | **73.5** | 78.5 | ✗ | ✗ | ✗ |
| Graph2SMILES (D-GAT) (*ours*) | 51.2 | 66.3 | 70.4 | 73.9 | ✗ | ✗ | ✗ |
| Graph2SMILES (D-GCN) (*ours*) | **52.9** | 66.5 | 70.0 | 72.9 | ✗ | ✗ | ✗ |
| MEGAN (Sacha et al., 2021) | 48.1 | 70.7 | 78.4 | 86.1 | ✗ | ✓ | ✗ |
| G2Gs (Shi et al., 2020) | 48.9 | 67.6 | 72.5 | 75.5 | ✗ | ✓ | ✗ |
| RetroXpert (Yan et al., 2020) | 50.4 | 61.1 | 62.3 | 63.4 | ✗ | ✓ | ✓ |
| GTA (Seo et al., 2021) | 51.1 | 67.6 | 74.8 | 81.6 | ✗ | ✓ | ✓ |
| RetroPrime (Wang et al., 2021b) | 51.4 | 70.8 | 74.0 | 76.1 | ✗ | ✓ | ✓ |
| GLN (Dai et al., 2019a) | 52.5 | 69.0 | 75.6 | 83.7 | ✓ | ✓ | ✗ |
| Aug. Transformer (Tetko et al., 2020) | 53.2 | - | 80.5 | 85.2 | ✗ | ✗ | ✓ |
| LocalRetro (Chen & Jung, 2021) | 53.4 | **77.5** | **85.9** | **92.4** | ✓ | ✓ | ✗ |
| GraphRetro (Somnath et al., 2020) | 53.7 | 68.3 | 72.2 | 75.5 | ✗ | ✓ | ✗ |
| Chemformer (Irwin et al., 2021) | 54.3 | - | 62.3 | 63.0 | ✗ | ✗ | ✓ |
| EBM (Dual-TB) (Sun et al., 2020) | **55.2** | 74.6 | 80.5 | 86.9 | ✓ | ✓ | ✓ |

The second group of rows in Table 3 includes methods that use additional features or techniques. Graph2SMILES beats a number of methods in this group with a top-1 accuracy of 52.9 without using any of templates, atom mapping or output SMILES augmentation. EBM (Dual-TB) (Sun

et al., 2020) achieves the SOTA for top-1 accuracy, and LocalRetro (Chen & Jung, 2021) for top-3, 5 and 10 accuracies respectively. Both make use of templates and atom mapping, which seem to be empirically helpful for this small dataset. Similar to how Graph2SMILES can benefit from using atom mapping as discussed earlier, we can make use of templates for potential gain, e.g. by retaining only candidates with reaction templates that have been seen in the training set.

## 4.5 ABLATION STUDY

We perform an ablation study to explore the effects of various components of Graph2SMILES by removing the D-MPNN, positional embedding, or the global attention encoder. We summarize the results on USPTO_50k, for which the quantitative effect is most conspicuous. From the results in Table 4, the removal of D-MPNN decreases the top-1 accuracy to 49.9. Note that this setup is architecturally similar to the Transformer baseline, but uses atom features rather than SMILES sequence as inputs. Getting rid of the graph-aware positional embedding leads to a drop in top-1 accuracy of 2.1 points for D-GCN and 0.6 points for D-GAT. The effect of removing the global attention encoder is more significant, decreasing the top-1 accuracy by up to 8.3 points. We therefore conclude that all three components of our encoder in Graph2SMILES, namely the D-MPNN, the graph-aware positional embedding, and the global attention encoder, are important. They work collectively to capture a meaningful representation of the input molecular graphs.

Table 4: Ablation study for Graph2SMILES on USPTO_50k.

| Architecture | Top-1 | Top-3 | Top-5 | Top-10 |
|---|---|---|---|---|
| Transformer baseline, AutoSynRoute (Lin et al., 2020) | 43.1 | 64.6 | 71.8 | 78.7 |
| Graph2SMILES without D-MPNN | 49.9 | 67.1 | 71.7 | 75.3 |
| Graph2SMILES (D-GCN) | 52.9 | 66.5 | 70.0 | 72.9 |
|     no graph-aware positional embedding | 50.8 | 65.2 | 69.4 | 73.6 |
|     no global attention encoder | 44.6 | 60.7 | 65.2 | 69.6 |
| Graph2SMILES (D-GAT) | 51.2 | 66.3 | 70.4 | 73.9 |
|     no graph-aware positional embedding | 50.6 | 65.7 | 69.8 | 73.2 |
|     no global attention encoder | 44.9 | 59.3 | 64.1 | 68.0 |

## 5 DISCUSSION

Throughout our experiments, to demonstrate the advantage over a vanilla Transformer, we have focused on the baseline Graph2SMILES model, forgoing the benefits of using additional features and techniques for performance engineering. Although we have used the top-1 accuracy as a basis for comparison throughout our discussion, we recognize its limitations especially for retrosynthesis, which can have many equally plausible options. Similarly, the datasets we use for reaction outcome prediction are not perfectly detailed, with legitimate ambiguity in the identity of the major product. While Graph2SMILES shows strong performance for top-1 accuracy and can replace Transformer model with minimal modification to the pipeline, there could be cases when top-n accuracy is more relevant (e.g. in multi-step planning applications). In those cases, the aforementioned performance engineering techniques would be necessary to boost the top-n performance of Graph2SMILES.

## 6 CONCLUSION

In this paper, we present a novel Graph2SMILES model for template-free reaction outcome prediction and retrosynthesis. The permutation invariance of its D-MPNN and graph-aware positional embedding eliminates the need for any input-side SMILES augmentation, while achieving noticeable improvement over the Transformer baselines, especially for top-1 accuracy. Graph2SMILES is therefore an attractive drop-in replacement for any methods that use the Transformer model for molecular transformation tasks. Further gain may be possible through performance engineering tricks that are orthogonal to the architecture itself, which will be investigated in future work.

REPRODUCIBILITY STATEMENT

We hereby declare that all of our reported results are reproducible from our existing code base, subject to minor deviations due to hardware-level numerical uncertainties as we have observed for some GPU models (e.g. V100). We include part of our code to reproduce some specific results, with a self-explanatory README file in the supplementary materials. We will open-source all scripts to reproduce any result in the manuscript along with pretrained Graph2SMILES checkpoints after the review period, regardless of whether the manuscript is accepted.

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

# A APPENDIX: ATOM AND BOND FEATURES USED

Table 5 summarizes the atom and bond features used in Graph2SMILES. Most features were adapted from GraphRetro (Somnath et al., 2020), with the addition of chiral features (R/S and E/Z).

Table 5: Atom and bond features.

| Feature | Possible values | Size |
|---|---|---|
| **Atom Feature** | | |
| Atom symbol | C, N, O etc. | 65 |
| Degree of the atom | $\{d \in \mathcal{Z}; 0 \leq d \leq 9\}$ | 10 |
| Formal charge of the atom | $\{d \in \mathcal{Z}; -2 \leq d \leq 2\}$ | 5 |
| Valency of the atom | $\{d \in \mathcal{Z}; 0 \leq d \leq 6\}$ | 7 |
| Hybridization of the atom | $sp, sp^2, sp^3, sp^3d, sp^3d^2$ | 5 |
| Number of associated hydrogens | 0, 1, 3, 4, 5 | 5 |
| Chirality | R, S, unspecified | 3 |
| Part of an aromatic ring | True, false | 2 |
| **Bond Feature** | | |
| Bond type | Single, double, triple, aromatic, other | 5 |
| Cis-trans isomerism | E, Z, unspecified | 3 |
| Conjugated | True, false | 2 |
| Part of a ring | True, false | 2 |

# B APPENDIX: OTHER METHODS FOR REACTION OUTCOME PREDICTION

Table 6: Results for reaction outcome prediction on USPTO_480k_separated for methods excluded in Section 4.3, sorted by top-1 accuracy. Best results are highlighted in **bold**.

| Methods | Top-$n$ accuracy (%) | | |
|---|---|---|---|
| | 1 | 3 | 5 |
| WLN/WLDN (Jin et al., 2017) | 79.6 | 87.7 | 89.2 |
| Seq2Seq (Schwaller et al., 2018) | 80.3 | 86.2 | 87.5 |
| GTPN (Do et al., 2019) | 83.2 | 86.0 | 86.5 |
| WLDN5 (Coley et al., 2019) | 85.6 | 92.8 | 93.4 |
| GRAT (Yoo et al., 2020) | 88.3 | - | - |
| Symbolic (Qian et al., 2020) | 90.4 | 94.1 | 95.0 |
| NERF (Bi et al., 2021) | **90.7** | 93.3 | 93.7 |
| Molecular Transformer (Schwaller et al., 2019) | 90.4 | **94.6** | **95.3** |

Table 6 summarizes the results for methods excluded in Section 4.3 for reaction outcome prediction, most of which report the values on the less challenging USPTO_480k_separated dataset, in which the reagents have been heuristically separated from the reactants. Only NERF shows marginal improvement of 0.3 points for top-1 accuracy over Molecular Transformer, whereas all other methods cannot perform as well. We therefore use Molecular Transformer as our baseline in Section 4.3. Note that ELECTRO (Bradshaw et al., 2019) tests on a simpler subset of reactions with linear electron flow (LEF), and we therefore exclude it from the quantitative comparison.

# C APPENDIX: SUMMARY OF FOUR USPTO DATASETS USED

Table 7: Statistics of USPTO datasets used.

| Dataset | Source | Train size | Validation size | Test size |
| --- | --- | --- | --- | --- |
| USPTO_480k_mixed | MT repo† | 409,035 | 30,000 | 40,000 |
| USPTO_STEREO_mixed | MT repo | 902,581 | 50,131 | 50,258 |
| USPTO_50k | GLN repo‡ | 40,008 | 5,001 | 5,007 |
| USPTO_full | GLN repo | 810,496 | 101,311 | 101,311 |

†https://github.com/pschwllr/MolecularTransformer ‡https://github.com/Hanjun-Dai/GLN.

# D APPENDIX: HYPERPARAMETER SETTING

Table 8: Hyperparameter setting used in the experiments for different datasets. Best settings selected based on validation are highlighted in **bold** if multiple values have been experimented.

| Dataset | Parameter | Value(s) |
| --- | --- | --- |
| All | Embedding size | **256**, 512 |
| | Hidden size (same among all modules) | **256**, 512 |
| | Filter size in Transformer | 2048 |
| | Number of D-MPNN layers | 2, **4**, 6 |
| | Number of D-GAT attention heads | 8 |
| | Attention encoder layers | 4, **6** |
| | Attention encoder heads | 8 |
| | Decoder layers | 4, **6** |
| | Decoder heads | 8 |
| | Number of accumulation steps | 4 |
| USPTO_480k | Batch type | Source token counts |
| | Batch size | 4096 |
| | Total number of steps | 300,000 |
| | Noam learning rate factor | 2 |
| | Dropout | 0.1 |
| USPTO_STEREO | Batch type | Source token counts |
| | Batch size | 4096 |
| | Total number of steps | 400,000 |
| | Noam learning rate factor | 2 |
| | Dropout | 0.1 |
| USPTO_50k | Batch type | Source token counts |
| | Batch size | 4096 |
| | Total number of steps | 200,000 |
| | Noam learning rate factor | 2, **4** |
| | Dropout | 0.1, **0.3** |
| USPTO_full | Batch type | Source+target token counts |
| | Batch size | 8192 |
| | Total number of steps | 400,000 |
| | Noam learning rate factor | 2 |
| | Dropout | 0.1 |

# E  APPENDIX: EFFECTIVENESS OF ADDITIONAL FEATURES OR TECHNIQUES

Table 9: Published results on USPTO_50k without reaction type, that demonstrate the effectiveness of additional features or techniques for the Transformer model and variants

| Method | Top-1 | Top-3 | Top-5 | Top-10 |
|---|---|---|---|---|
| EBM reranking using Transformer (Sun et al., 2020) | | | | |
|     on template-free proposal | 53.6 | 70.7 | 74.6 | 77.0 |
|     on proposal constrained by templates | **55.2** | **74.6** | **80.5** | **86.9** |
| GTA, non-augmented (Seo et al., 2021) | | | | |
|     without cross-attention | 46.8 | 65.2 | 70.5 | 74.9 |
|     with cross-attention (using atom-mapping) | **47.3** | **66.7** | **72.3** | **76.5** |
| DMP, with pretrained Transformer encoder (Zhu et al., 2021) | | | | |
|     Transformer baseline | 42.3 | 61.9 | 67.5 | 72.9 |
|     fusion with pretrained ChemBERTa encoder | 43.9 | 62.2 | 68.0 | 73.1 |
|     fusion with pretrained DMP encoder | **46.1** | **65.2** | **70.4** | **74.3** |
| Augmented Transformer (Tetko et al., 2020) | | | | |
|     x20 augmentation for products only | 42.5 | - | - | - |
|     x20 augmentation for both products and reactants | **48.0** | - | - | - |
| SCROP (Zheng et al., 2020) | | | | |
|     Transformer baseline | 43.3 | 59.1 | 64.0 | 67.0 |
|     reranked with syntax corrector | **43.7** | **60.0** | **65.2** | **68.7** |
| Latent Transformer, non-augmented (Chen et al., 2020) | | | | |
|     no latent variable (N=1) | 42.0 | 57.0 | 61.9 | 65.7 |
|     with latent variable (N=2) | **42.1** | **60.0** | **64.9** | **70.3** |

For all groups of rows in Table 9, the first row of numbers are for the Transformer variants where features or techniques of interest are not used, and the rest of the rows otherwise. In turn, these results illustrate the effectiveness of using templates (EBM), atom-mapping (GTA), pretraining (DMP), output-side augmentation (Augmented Transformer), syntax-based reranking (SCROP) and variational inference (Latent Transformer). Inclusion of these features or techniques clearly improves the accuracies across the board over respective baselines, as highlighted in **bold**.

Table 10: Graph2SMILES results on USPTO_50k without reaction type with latent variables

| Model | Top-1 | Top-3 | Top-5 | Top-10 |
|---|---|---|---|---|
| Graph2SMILES (D-GCN) | | | | |
|     no latent variable (N=1) | **52.9** | 66.5 | 70.0 | 72.9 |
|     with latent variable (N=2) | 52.0 | **70.2** | **75.2** | **79.5** |
| Graph2SMILES (D-GAT) | | | | |
|     no latent variable (N=1) | **51.2** | 66.3 | 70.4 | 73.9 |
|     with latent variable (N=2) | 50.3 | **68.8** | **73.7** | **77.7** |

As an illustration of how Graph2SMILES can benefit from additional features or techniques, we briefly experiment with one of them, latent variable modeling similar to Chen et al. (2020) and Kim et al. (2021), using $N = 2$ latent classes with a uniform prior. As in Table 10, this noticeably increases the top-n accuracies for both D-GCN and D-GAT, which can be particularly relevant for multi-step planning.

# F   APPENDIX: VISUALIZATION OF CASES WHERE GRAPH2SMILES OUTPERFORMS THE TRANSFORMER BASELINE

We include some real cases for which Graph2SMILES gives the correct top-1 prediction but the Molecular Transformer baseline cannot, for the reaction outcome prediction task on USPTO_480k_mixed. For each row in Figure 2, the reactants are shown on the left, the ground truth product (which is also the correct prediction by Graph2SMILES) is shown in the center, and the erroneous prediction by the Molecular Transformer baseline is shown on the right.

Figure 2: Real cases for which Graph2SMILES predicts the major product correctly on USPTO_480k_mixed. Top 2 rows: N-capping followed by intramolecular ring closing. Third row: deamination. Fourth row: regioselective substitution.

The first two rows represent two intramolecular ring forming reactions after an N-alkylation or amidation. As shown by the right-most molecules, without explicit knowledge of the reactant graphs, Molecular Transformer instead predicts substitution reactions at the benzylic carbon and at the phenyl carbon respectively. Both predictions seem to be based solely on local chemical environments, with the first being plausible as nucleophilic substitution on benzyl bromide, but the second nonsensical since such electrophilic aromatic substitution would not happen with the nucleophilic phenylamine. Graph2SMILES, on the other hand, correctly predicts the ring formation. We therefore hypothesize that graph representation may be more powerful at encoding beyond local contexts via its global attention mechanism, which is potentially important for such reactions. Further quantitative evaluation is left as future work, since explainability is not a trivial task for template-free models.

The third row represents the removal of the amine group attached to the benzene ring, for which Graph2SMILES gives the correct prediction, but the Molecular Transformer predicts the formation of an aryl hydrazine, possibly by detecting the extra nitrogen in the input SMILES (from DMF) but failing to realize it as merely a spectator solvent without full recognition of its graph structure. The last row is a case where Graph2SMILES predicts the correct regioselectivity (ortho vs. para to the nitro group) for the aromatic substitution reaction. This may be interpreted as better capability of Graph2SMILES to learn longer range topological information as compared to the Transformer baseline.

