# OpenReview forum: "Permutation invariant graph-to-sequence model for template-free retrosynthesis and reaction prediction"
_ICLR.cc/2022/Conference — ICLR 2022 Submitted_

### Official Review · Reviewer_Wxfa · 2021-11-01

**Correctness:** 4
**Technical Novelty And Significance:** 2
**Empirical Novelty And Significance:** Not applicable
**Recommendation:** 5
**Confidence:** 4

**Main Review:**

(+)

Using a sequence-independent encoding in the easy-to-use transformer makes sense and is a research question which is interesting for the - I think mostly, AI in chemistry - community.

(-)

- As far as I can see, the technical novelty is limited. The proposal is a combination of rather well-known methods.

 D-MPNN (where attention is added) and

re-parameterized, relative positional encodings

 (using 0 to represent atoms in different molecules)

in transformers

- It is unclear if the proposed attention in the GNN is useful: Since we only see "For reaction outcome prediction, there is a small advantage of using D-GAT over D-GCN", and the ablation results are missing.

- The results in Table 1 are only convincing for USPTO_STEREO_mixed.

- The experimental comparison for retrosynthesis compares to methods which use different forms of pretraining or augmentation, which makes sense. It shows that the augmentation still provides advantages. And, as far as I understand, the graph-based molecule embedding entails that Smiles augmentation would not improve your results further? Also, it would make sense to include the ablation results for Graph2Smiles just using the transformer into Table 2 directly.


-------------------------------
Other Comments


- It is unclear to me what "Our hyperparameters for D-GAT and D-GCN are adapted from GraphRetro" means.
- In (8), s_uv should be AttnSum(...)
- How exactly is \mathcal{B}_u,v used in the learnable \tilde{r}_u,v?
- Table 4:
Since the paper proposes the attention-based GNN, the ablation should be provided for that model.
- Table 4: What is "no global attention encoder"? Just the combination of GNN embeddings w/o transformer? Since transformer is the baseline, I would not consider this as an ablation setting.
- Table 4: How do the results look on the other tasks? For retrosynthesis, all open existing systems that yield full retrosynthesis trees which I know use top50 (or similar), so the fact that the base model is better at top10 (already) renders the analysis questionable.
- "We include part of our code to reproduce some speciﬁc results" - Why not all?


**Summary Of The Paper:**

The paper proposes a GNN-based extension of transformers, which have been shown to be effective for reaction prediction etc. before. In particular, the GNN-based embedding of molecules in the reaction embeddings overcomes the artificial bias inherent in the often applied sequence embeddings. The experiments show that there are sometimes increases of performance in reaction and retrosynthesis prediction - and the approach could be applied to similar problems.




**Summary Of The Review:**

Overall, I think the authors' proposed model for reaction prediction makes sense. However, as mentioned above, the paper's writing could be improved, the technical contribution is limited, and the experiments also show only limited improvements. Altogether, I therefore suggest to reject the paper at the current moment. I am happy to adjust my score in case I missed critical parts.

---

> ### Author Response · Authors · 2021-11-21
> **Reply to R4 (Part 1)**
>
> We thank the reviewer for the feedback and comments. We address each of them in turn (with some reorganization).
>
> **Q1. “The technical novelty is limited”.**
> Please see our response to all reviewers with the title “Reply to R1, R2, R3, R4: technical contribution is incremental”.
>
> **Q2. Comments related to the main results.**
> **2a. “Unclear if the proposed attention in the GNN is useful”**
> We consider D-GAT and D-GCN simply to be design choices for the graph encoder rather than a major contribution of this work. We have updated Figure 1 as well as relevant sentences in Sections 1 and 2 to make this more explicit. We would like to point out that D-GAT also yields “improvement on top-5 and 10 accuracies for USPTO_50k as in Table 3”. Nevertheless, the little difference between the performances of the D-GAT and the D-GCN variants makes it sensical to leave both variants as possible design choices, both of which we have provided sample codes for and have included in the open-source release.
>
> **2b. “The results in Table 1 are only convincing for USPTO_STEREO_mixed”**
> Graph2SMILES shows similar improvement over Molecular Transformer baseline for USPTO_480k_mixed for all top-n accuracy. We argue that the relative advantage of our architecture over pure Transformer is not rendered less valid despite the small gap to Augmented Transformer [1] and Chemformer [2]. Augmented Transformer uses x100 SMILES augmentation and Chemformer uses pretraining with a much larger model. Both methods are significantly more expensive than the Molecular Transformer or Graph2SMILES; in particular, x100 test time SMILES augmentation makes the Augmented Transformer impractical to deploy in high-throughput predictive chemistry workflows.
>
> **2c. “Include the ablation results for Graph2Smiles just using the transformer into Table 2 directly.”**
> Thanks for the reviewer’s suggestion. We have revised Table 2 to include the Transformer baseline from a new reference [3] that another reviewer points to.
>
> **2d. “For retrosynthesis” ... “the base model is better at top10”... “renders the analysis questionable”.**
> For retrosynthesis and other recommender tasks, the tradeoff between top-1 and top-n accuracy is not unexpected. Most notably, from Table 3, the top-10 performance of the Transformer baseline (AutoSynRoute) is at 78.7%, higher than quite a few of its, supposedly improved variants (GET, DMP fusion [3], Tied Transformer, RetroXpert, RetroPrime, Chemformer). This phenomenon makes sense because the training objective is only related to the top-1 prediction, but has not been rigorously explained otherwise (except briefly without justification in EBM [4] and Chemformer to the best of our knowledge). We therefore believe that better performance of the baseline at top-10 would not render the analysis of our work (and of the aforementioned previous works) questionable, since we prioritize top-1 accuracy in training and validation in a similar way.
>
> For practical use in multi-step planning, we agree with the reviewer that top-n accuracy might be more relevant (as in Section 5). We believe that this can be improved by using additional established performance engineering techniques. We include an additional Appendix E in our revised submission to demonstrate the quantitative effects which have been reported in previous studies. Further, we incorporate one such performance engineering technique, latent variable modeling, in a new experiment in  Appendix E as empirical evidence that top-n accuracies can be improved. From Table 10, using latent classes N = 2 can already boost the top-10 accuracies of the D-GCN variant from 72.9 to 79.5, and of D-GAT from 73.9 to 77.7.
>
> **2e. “Augmentation still provides advantages” but “would not improve your results further”.**
> Note that in Table 2 and 3, the last column indicates whether *output* data augmentation is used. The permutation invariance of Graph2SMILES indeed means that it would not benefit from input-side SMILES augmentation. However, with the autoregressive loss, the training process is not invariant to changes in the target SMILES. As such, Graph2SMILES might still benefit from output-side SMILES augmentation, as has been demonstrated by Augmented Transformer. In Table 9 of our newly included Appendix E, output (i.e. reactant) SMILES augmentation is able to increase the top-1 accuracy of Augmented Transformer to 48.0 on USPTO_50k, from 42.5 without output augmentation.
>
> We leave the quantification of effects from output-side SMILES augmentation on Graph2SMILES as future work (along with the empirical study of including other features/techniques).

---

> > ### Author Response · Authors · 2021-11-21
> > **Reply to R4 (Part 2)**
> >
> > **Q3. Comments related to the ablation study.**
> > Firstly, we would like to clarify that our ablation study in Section 4.5 mainly aims to quantify the effect of various components (D-MPNN, graph-aware positional embedding and global attention encoder) of the graph encoder in Graph2SMILES. We therefore ablate graph-aware positional embedding and global attention encoder sequentially in Table 4 in the original submission, and have included the effect of ablating the D-MPNN in Table 4 in our revision. Regarding specific comments,
> >
> > **3a. “The ablation should be provided for that model.” (i.e. D-GAT). “The ablation results are missing” for D-GAT over D-GCN**
> > We consider D-GAT and D-GCN more as two design variants rather than ablation settings, and have included the results for both through Tables 1-3. We appreciate the reviewer’s suggestion and have updated Table 4 to include the ablation for both D-GCN and D-GAT variants for completeness. The quantitative effects of three components on top-1 accuracy are consistent for both variants.
> >
> > **3b. “Table 4: What is "no global attention encoder"? Just the combination of GNN embeddings w/o transformer? Since transformer is the baseline, I would not consider this as an ablation setting.”**
> > Yes, it refers to the combination of GNN + Transformer decoder, without the Transformer encoder. We see this as an ablation setting since the Transformer encoder is ablated but the other elements of the GNN encoder and Transformer decoder are kept.
> >
> > **Q4. “It is unclear to me what "Our hyperparameters for D-GAT and D-GCN are adapted from GraphRetro" means”; “In (8), s_uv should be AttnSum(...)”**
> > Thanks for pointing these out. We have revised the text to provide additional clarity and corrected this typo.
> >
> > **Q5. “How exactly is \mathcal{B}_u,v used in the learnable \tilde{r}_u,v?”**
> > \mathcal{B}_u,v is used in a similar way as a token index, to look up the respective \tilde{r}_u,v vector in the embedding matrix.
> >
> > **Q6. "We include part of our code to reproduce some speciﬁc results - Why not all?”**
> > This statement only applied to the sample codes we provided as supplementary materials at the time of submission, in which we also included preprocessed data to facilitate reproduction, but only for the small USPTO_50k dataset due to file size constraint. In our open-source version on GitHub, all codes have been included with more streamlined scripts for fetching the data from the server (which may otherwise break anonymity).
> >
> > [1] Tetko et al. State-of-the-art augmented NLP transformer models for direct and single-step retro synthesis. Nature Communications 2020.
> > [2] Irwin et al. Chemformer: A pre-trained transformer for computational chemistry. DOI: 10.33774/chemrxiv-2021-v2pnn.
> > [3] Zhu et al. Dual-view molecule pre-training. https://arxiv.org/abs/2106.10234.
> > [4] Sun et al. Towards understanding retrosynthesis by energy-based models. NeurIPS 2021.

---

> > > ### Comment · Reviewer_Wxfa · 2021-11-24
> > > **Response**
> > >
> > > Thank you for the detailed replies. As I wrote originally, the paper is generally good in my opinion.
> > >
> > > However, I do not think it fits into ICLR:
> > > - In view of the related work on graph-transformers it is a somewhat straightforward adaptation of transformers to molecules.
> > > - I do not think the architecture will have great impact beyond the transformers-on-molecules field, and so far transformers are not even used in the open source "full retrosynthesis" systems (i.e., which beyond a single step return the full path) but they use template-based models.
> > > - The experiments are not impressive either. The ablation is done on a single dataset and more interesting questions are not investigated, but your reply was ".. the question remains largely unresolved in the field". In particular, the fact that you emphasize the model increases top-1 while top-10 is decreased is critical. Recently there have been a couple of papers arguing that optimizing for top-1 alone is not effective for finding the full paths.

---

> > > > ### Author Response · Authors · 2021-11-29
> > > > **Clarifications for R4**
> > > >
> > > > We respect the reviewer’s decision to keep the score as-is and appreciate the reviewer for bringing up the relevance to full-path planning. We would like to clarify a few additional points, and offer alternative, if not complementary viewpoints on the full-path problem.
> > > >
> > > > **Re: scope of Graph2SMILES**
> > > > For one-step retrosynthesis, it is our intention to focus the scope on empirical improvement, similar to previous works accepted ML venues [1]-[6]. We are happy to offer clarifications within this scope, but we do not have rigorous answers for out-of-scope questions. Our reply that “the question remains largely unresolved in the field” is specifically answering Reviewer 1’s question of “why our method, or template-free methods in general can sometimes work better than template-based methods”. This is *not* a general answer to any questions unexplored by Graph2SMILES.
> > > >
> > > > **Re: full path planning**
> > > > Firstly, we would like to highlight that evaluations based top-1 and top-n accuracies have been a standard for one-step retrosynthesis, especially at ML venues [1]-[6]. While we respect the reviewer’s perspective that optimizing for top-1 alone may not be effective in the context of full path planning, we think it is still highly relevant. For example, in a general MCTS setting, it can be a good surrogate model at the rollout phase. Since the model is applied recursively, possibly solely on top-1 suggestion in some settings, a high top-1 accuracy can potentially lead to much more accurate reward and hence enhance overall performance.
> > > >
> > > > To the best of our knowledge, at least rxn4chemistry [7] uses the Transformer model as their one-step proposer. We agree with the reviewer that full retrosynthesis systems tend to use template-based models, but we do not see it as a ground for discrediting non template-based models. In fact, we believe that part of the reason why translation-based approaches have not been widely adapted for multi-step is the practical complications due to the need for SMILES augmentation (especially during inference time), as we have discussed in our Introduction section. This is exactly the main motivation for Graph2SMILES -- to mitigate the need for data augmentation so that the pipeline can be simpler when Graph2SMILES is plugged in.
> > > >
> > > > [1] Dai et al. Retrosynthesis Prediction with Conditional Graph Logic Network. NeurIPS 2019.
> > > > [2] Shi et al. A Graph to Graphs Framework for Retrosynthesis Prediction. ICML 2020.
> > > > [3] Yan et al. Retroxpert: Decompose retrosynthesis prediction like a chemist. NeurIPS 2020.
> > > > [4] Sun et al. Towards understanding retrosynthesis by energy-based models. NeurIPS 2021.
> > > > [5] Somnath et al. Learning graph models for template-free retrosynthesis. NeurIPS 2021.
> > > > [6] Seo et al. GTA: Graph truncated attention for retrosynthesis. AAAI 2021.
> > > > [7] https://github.com/rxn4chemistry/rxn4chemistry.

---

> > > > > ### Comment · Reviewer_Wxfa · 2021-11-30
> > > > > **Response**
> > > > >
> > > > > Thank you for the detailed reply.
> > > > >
> > > > > - **Top-1** \
> > > > > The aspect with the surrogate model makes sense. The fact that the ML venues focus on top-n is true but also questionable.
> > > > >
> > > > > - **No Discredits** \
> > > > > I definitely agree that using template-free models would be more interesting; I know about rxn4chemistry and therefore mentioned *open-source* tools. The "(especially during inference time)" might be the really interesting aspect and should be stressed more, and maybe shown experimentally in a scenario with same pretraining. Augmentation at training time is not really an issue (and yields good performance) if I understand correctly.
> > > > >
> > > > > As I wrote above, I recommend to reject the paper because I do not think that the proposal/findings are interesting enough or will have enough impact on AI research, the primary area of ICLR. Furthermore, the discussion has highlighted parts of the experiments that do not fully show the model's impact - no need for augmentation at inference time. Nevertheless, the last reply of the authors has highlighted some aspects which I think are important and I will take account of them in the score.

---

### Official Review · Reviewer_6G2y · 2021-11-01

**Correctness:** 2
**Technical Novelty And Significance:** 2
**Empirical Novelty And Significance:** 2
**Recommendation:** 3
**Confidence:** 4

**Main Review:**

The proposed framework integrates several recently developed engineering techniques and empirically shows its superior performance over vanilla SMILES-to-SMILES transformer baseline. This paper provides another comparison baseline for research on retrosynthesis and reaction prediction. Nevertheless, I have the following concerns regarding the paper.

1.	The proposed framework is similar to the NERF approach (Bi et al. ICML 2021) as cited by the authors. NERF formulates the reaction prediction problem as a graph-to-graph translation problem. Also, NERF first leverages graph neural networks to capture the local information in individual molecules and then utilizes a Transformer encoder to further models the intermolecular interactions between nodes from multiple molecules. Furthermore, NERF uses a Transformer decoder to decode the output as graph. These are almost the same as that in the method proposed in the paper. The only different to me is that the NERF uses a Transformer to decode the output into graph directly (in a non-autoregressive fashion), while the proposed method here uses the Transformer to decode the output into SMILES strings (in an autoregressive fashion). In this sense, the novelty of this paper is limited to me. Note: I think NERF can naturally apply to two or more molecules since the Transformer encoder is used by considering all node embeddings from multiple molecules as a node set.
2.	Experimentally, the proposed method is not directly compared with NERF. I think such comparison is necessary since as shown in the paper, NERF outperforms the SMILES-to-SMILES transformer baseline, and even the augmented version of it, to which the proposed method here obtained inferior performance. The two methods are similar and closely related. I would expect the paper to include NERF into the main results in Table 1. Also, for the USPTO_STEREO_mixed task, I wonder what the reason was for only comparing with the vanilla Transformer, and why not comparing with the Augmented Transformer or the state-of-the-art method Chemformer?
3.	Results in Table 1 show that, the proposed method is inferior to the Transformer baseline with simple augmentation. Also, 1% less than the tested method Chemformer, which makes the paper’s contribution less significant to me.
4.	The claim in the Abstract “molecular graph encoders that mitigates the need for input data augmentation” is a strong claim to me. Nevertheless, there is no evidence to support that claim. The slightly better performance over the SMILES-to-SMILES transformer baseline is not a convincing evidence to me. Input data augmentation may play a significant role on regularizing the deep neural networks. I think better justification to support the claim is necessary.
5.	The statement in the last sentence of the first paragraph on Page2:  “[SMILES augmentation]…be interpreted as evidence of the ineffectiveness of the SMILES representation itself.” I think this hypothesis may need better support and analysis. To me, the augmentation of SMILES strings can act as a model regularization method, which helps the trained model to generalize well to unseen data, and may not directly infer the ineffectiveness of the SMILES representation itself.
6.	I am not fully understand the claim in the second paragraph of Page2 “… we guarantee the permutation invariance of Graph2SMILES to the input, eliminating the need for input-side augmentation altogether.” I think it would be useful to specify how and why so.
7.	The proposed method integrates several performance engineering techniques (such as attention weights and multi-headed attention in the graph encoder, integration of shortest path length in the positional embeddings etc.), so where the improvement is really coming from is not clear to me. In the ablation study in Table4, both the positional embedding and global attention are key to the Transformer’s performance, so the performance degradation is expected when remove them: Transformer expects a positional embedding to work, and without a global attention the encoder will not be able to capture information from multiple molecules (their graphs are disconnected).


**Summary Of The Paper:**

This paper proposes a graph-to-SMILES framework, which incorporates several recently developed engineering techniques from the community, for synthesis planning and reaction outcome prediction tasks. The proposed method leverages graph neural networks and Transformer attention model to encode the graph inputs and then utilizes a Transformer decoder to generate the SMILES string as outputs. Experiments on benchmark retrosynthesis and reaction prediction tasks show that the proposed approach outperformed the vanilla SMILES-to-SMILES transformer baseline, but obtained inferior results than some other advanced methods. The paper is interesting, but both the technical novelty and the experimental studies are weak to me.

**Summary Of The Review:**

The proposed method is similar to NERF as proposed by Bi et al., so the technical novelty is limited. Also, the experiment study missed important comparison baselines. Furthermore, a more comprehensive ablation study is needed since several engineering techniques are employed, and it is difficult to tell where the performance improvement is really coming from when compared to the Transformer vanilla model.

---

> ### Author Response · Authors · 2021-11-21
> **Reply to R3 (Part 1)**
>
> We thank the reviewer for the feedback and comments. We address each of them in turn.
>
> **Q1. “The proposed framework is similar to the NERF approach (Bi et al., 2021)”.**
> There are a few crucial differences between NERF [1] and our work. Most importantly, NERF makes a strong assumption that the input and output graphs share the same set of atoms. This assumption is permissible for reaction prediction but not for retrosynthesis, where there will be heavy atoms present in the output reactant graph that do not appear in the input product graph (e.g., atoms in leaving groups). The NERF formulation as described by Bi et al. is fundamentally unable to be applied to retrosynthesis. Our formulation, however, does not rely on this assumption. The same Graph2SMILES model works for both forward prediction and retrosynthesis, and is therefore more flexible.
>
> Secondly, NERF uses a Conditional VAE formulation that maximizes the ELBO, whereas we maximize the conditional likelihood itself (Eqn 13). Because of the CVAE formulation, NERF requires the product molecular graph as an input during training, in order to encode the posterior $q(z|G^p, G^r)$ where $G^p$ and $G^r$ are the product and reactant graphs respectively. More specifically, this posterior is modeled using a Transformer-Decoder with both $G^p$ and $G^r$ as inputs. When Graph2SMILES is used for reaction prediction, however, it does not need $G^p$ at all. Neither does Graph2SMILES have the Transformer-Decoder component in its graph encoder like NERF does. We have revised the manuscript to include this important architectural difference between Graph2SMILES and NERF in Section 3.3.
>
> Lastly, there are other differences in both the encoder and the decoder. Our D-GAT/D-GCN is different from GCN or non-directed MPNN in general as we explain in Section 2.2.1. The graph positional embedding also contributes to the performance gain as we show in Section 4.5. Without positional embedding or extra topological features, a vanilla GCN+Transformer encoder would have limited expressivity, since it cannot differentiate between certain molecules (e.g. cyclohexane vs. 2 cyclopropanes, or bi-cyclopentane vs. decalin). As for the decoder, NERF indeed uses a Transformer before PointNets in their decoder, but we view its purpose more as additional encoding layers after incorporating the latent $z$; in fact, this part is named “Transformer-Encoder” in their Eqn (9). Our Transformer decoder, in contrast, only attends to the $h^r$ from encoder output through encoder-decoder attention, without updating $h^r$ any further.
>
> **Q2 and Q3. “not directly compared with NERF … Augmented Transformer or … Chemformer” and “inferior to the Transformer baseline with simple augmentation”**
> It is our intention to only compare with published results. To the best of our knowledge, Augmented Transformer and Chemformer do not evaluate on USPTO_STEREO; NERF only reports on the less challenging USPTO_480k_separated (referred to as USPTO-MIT in their paper) in which the reagents have been separated from the reactants using knowledge of the ground truth product. We choose not to use techniques such as pretraining in Chemformer in order to focus on the effectiveness of the Graph2SMILES architecture itself as a drop-in replacement for the Transformer.
>
> Note that our results for “Augmented Transformer” are from [2], which uses x100 SMILES augmentation during both training and testing, followed by heuristic-based ensembling and deduplication. We do not see it as “simple augmentation” since it has a much more complicated and costly model pipeline. There may be some nomenclature confusion with “Transformer-augmented” in Table 1 of the NERF paper, which only uses x2 SMILES augmentation during training. The baseline results we report for “Molecular Transformer'' in our Tables 1 and 6 correspond to NERF’s “Transformer-augmented”, which our method does outperform.

---

> > ### Author Response · Authors · 2021-11-21
> > **Reply to R3 (Part 2)**
> >
> > **Q4, Q5, Q6**
> > We address these questions together with a 3-part response.
> > **1) “Permutation invariance of Graph2SMILES to the input”**
> > A SMILES string is a linearized representation of the molecular graph. With toolkits such as RDKit, graphs reconstructed from equivalent SMILES would be exactly the same and invariant to how the SMILES is written. Atom ordering or atom indices are not used in the Graph2SMILES pipeline at any stage. Graph2SMILES is therefore permutation invariant to the input as well, since each of the D-MPNN, attention encoder, and Transformer decoder guarantees such invariance by design.
> >
> > **2) “Mitigates the need for input data augmentation”**
> > This claim directly follows from the permutation invariance to the input side and needs no proof. Since our molecular graph input will be exactly the same for equivalent SMILES, input side SMILES augmentation is equivalent to training with the original input graph repeatedly, and hence provides no benefit. We show that Graph2SMILES outperforms SMILES-to-SMILES empirically to complement rather than prove the claim.
> >
> > **3) “Not directly infer the ineffectiveness of the SMILES representation”**
> > The reviewer makes a good point about the potential regularization effect when generalizing to unseen data. We have revised the sentences accordingly in the latest version, which now discuss the practical complications of SMILES augmentation instead.
> >
> > **Q7. “Performance engineering techniques” make it difficult to tell where "improvement is really coming from"**
> > The effectiveness of attention weights and multi-headed attention in the graph encoder has been established, at least in GAT [3] and GATv2 [4]. It has also been common in recent works for retrosynthesis (e.g. GET [5] and RetroXpert [6]) and other molecular ML tasks [7][8].  We therefore consider D-GAT as a design variant, rather than a performance engineering technique. We have also quantitatively compared D-GAT and D-GCN in all of our main results in Tables 1-3.
> >
> > For the relative positional embedding, from our Eqn (12), the shortest path length is the only pairwise feature we use to parameterize the embedding. Without it, the second term in Eqn (12) would disappear and there would be no positional embedding at all. We therefore argue that it is a crucial component of our graph encoder rather than a performance engineering technique. The results without positional embedding are shown in the ablation study, as the reviewer recognizes.
> >
> > We appreciate and agree with the reviewer’s interpretation of Table 4. We agree that some performance degradation might be expected when ablating certain features of the Graph2SMILES model, but we believe there is still value in quantifying the extent of this degradation to confirm their importance.
> >
> > [1] Bi et al. NERF. ICML 2021.
> > [2] Tetko et al. State-of-the-art augmented NLP transformer models for direct and single-step retro synthesis. Nature Communications 2020.
> > [3] Veličković et al. Graph attention networks. ICLR 2018.
> > [4] Brody et al. How attentive are graph attention networks? https://arxiv.org/abs/2105.14491.
> > [5] Mao et al. Molecular graph enhanced transformer for retrosynthesis prediction. Neurocomputing 2021.
> > [6] Yan et al. Retroxpert: Decompose retrosynthesis prediction like a chemist. NeurIPS 2020.
> > [7] Hu et al. Strategies for pre-training graph neural networks. ICLR 2020.
> > [8] Li et al. An effective self-supervised framework for learning expressive molecular global representations to drug discovery. Briefings in Bioinformatics 2021.

---

> > > ### Comment · Reviewer_6G2y · 2021-11-24
> > > **Main concerns remain**
> > >
> > > I appreciate the authors' feedback on my initial reviews. My main concerns, however, remain.
> > >
> > > First, from the authors' answer to Q1, I still think that the two approaches (i.e., the proposed one and NERF) are similar. The authors highlight some similarities and dissimilarities between the two methods, and I think they are helpful. On the other hand, how those minor modifications made impact the performance of the proposed method is not clear to me, neither theoretically nor empirically.
> > >
> > > Second, I think a comprehensive comparison with Augmented Transformer, NERF, and Chemformer is necessary. I understand the authors' argument of “our intention to only compare with published results”, but the Augmented Transformer is the very basic baseline of the proposed work, the NERF method is technically similar to what is  proposed, and the Chemformer stands for the state-of-the-art strategy in the field. Without those comparisons, it is quite difficult for me to judge the contributions of this work.
> > >
> > > Third, I am not convinced by the authors' claim that “we guarantee the permutation invariance of Graph2SMILES to the input, eliminating the need for input-side augmentation altogether.” I agree with the first part of the claim, but not the second part. As highlighted in my initial reviews, this is a very strong claim, and I would expect better evidence to support that claim. In my opinion, input-side graph data augmentation has other effects including model calibration, adversarial attacks, over-smoothing, and over-squashing, etc, and I think the slightly better accuracy over the SMILES-to-SMILES transformer baseline is not a convincing evidence to me.

---

> > > > ### Author Response · Authors · 2021-11-29
> > > > **Clarifications for R3**
> > > >
> > > > We respect the reviewer’s decision to keep the score as-is, but would still like to clarify a few additional points.
> > > >
> > > > **Re: similarity with NERF**
> > > > As we explain in the previous reply, NERF uses the product graph not only as the target, but also as an input to its model. We therefore view NERF as significantly different from all other methods, including Graph2SMILES, that do not use the product graph as an input. Nevertheless, we respect the reviewer’s opinion.
> > > >
> > > > **Re: clarifications on input-side graph data augmentation**
> > > > We realize that the reviewer might have some confusion about what “input-side graph data augmentation” means in the context of reaction outcome prediction and retrosynthesis. The data augmentation we refer to and previously done by other works, is solely with multiple equivalent SMILES for the exact same reaction. The molecular identities are exactly the same. As a simple example from Augmented Transformer,
> > > >
> > > > *Original SMILES:*
> > > > CC(=O)c1ccc(Br)nc1.CNC>>CC(c1ccc(Br)nc1)N(C)C
> > > > *Augmented SMILES:*
> > > > CC(=O)c1ccc(nc1)Br.CNC>>CC(c1ccc(Br)nc1)N(C)C
> > > >
> > > > where the (Br) and (nc1) branches at the input side are swapped. The two SMILES, however, still have exact same input molecules (albeit written differently). Any permutation invariant graph encoder, like the one in Graph2SMILES, will therefore guarantee that atom embeddings coming out of the encoder will be exactly the same for the two SMILES inputs. It then follows that “input-side graph data augmentation” will not benefit Graph2SMILES; training with the original SMILES and the augmented SMILES is theoretically equivalent to training repeatedly with the original SMILES.
> > > >
> > > > The reviewer might have interpreted “graph data augmentation” differently, e.g. by perturbing the structures and augmenting with different molecules. This kind of augmentation, however, has never been done for these tasks to the best of our knowledge. Changing the molecular identities essentially "invents" new reactions, which is chemically nontrivial.

---

### Official Review · Reviewer_3c52 · 2021-11-02

**Correctness:** 3
**Technical Novelty And Significance:** 2
**Empirical Novelty And Significance:** 2
**Recommendation:** 6
**Confidence:** 4

**Main Review:**

The main strengths of this paper are as follows.
1.	This paper proposes Graph2SMILES, which is a graph-to-sequence architecture without using sequence representations of input SMILES. Therefore, Graph2SMILES is permutation invariant to the input and does not need the data augmentation.
2.	Graph2SMILES has a wide range of potential applications because it can serve as a drop-in replacement for Transformer in many tasks involving the molecule(s)-to-molecule(s) transformation.

My major concerns are as follows.
1.	This paper states that Graph2SMILES achieves state-of-the-art top-1 accuracy on common benchmarks among methods that do not use reaction templates, atom mapping, pretraining, or data augmentation strategies. The authors claim that integrating the above features or techniques with Graph2SMILES could improve the performance. However, they do not conduct experiments to demonstrate their claim. Besides, as the aforementioned techniques are commonly seen in predictive chemistry tasks, the authors may want to explain why they do not equip Graph2SMILES with these techniques.
2.	D-GAT is a variant of D-GCN with attention-based message updates. However, D-GAT does not outperform D-GCN in terms of the top-1 accuracy, which is the basis for comparison throughout the discussion in this paper. According to Table 1, D-GAT has a small advantage over D-GCN only in terms of the top-5 and top-10 accuracies in the reaction outcome prediction.
3.	Graph2SMILES involves calculating pairwise shortest path lengths between atoms, which can be computationally prohibitive. The authors may want to compare Graph2SMILES against baselines in terms of the computational complexity.



**Summary Of The Paper:**

This paper proposes a graph-to-sequence architecture called Graph2SMILES for the retrosynthesis and the reaction outcome prediction. Graph2SMILES uses an attention-augmented D-MPNN encoder to capture the local information and a global attention encoder with graph-aware positional embeddings to capture the global information. Experiments show that Graph2SMILES is competitive with Transformer baselines but does not outperform state-of-the-art methods on tasks of the one-step retrosynthesis and the reaction outcome prediction.

**Summary Of The Review:**

This paper studies two important problems in the computer-aided organic chemistry and proposes a graph-to-sequence architecture called Graph2SMILES. However, the empirical results do not show a superior performance of Graph2SMILES to existing methods, and the technical contribution is incremental.

---

> ### Author Response · Authors · 2021-11-21
> **Reply to R2**
>
> We thank the reviewer for the feedback and comments. We address each of them in turn.
>
> **Q1. Why “not equip Graph2SMILES with” … “features or techniques''**
> It is indeed our intention to mainly demonstrate Graph2SMILES as a new backbone and a drop-in replacement for the Transformer. To this end, we believe that it suffices to show its superior performance over the Transformer baselines in the absence of additional performance engineering techniques. In this way, we also keep the model simple and easily adaptable for future research efforts.
>
> We agree with the reviewer that a more quantitative demonstration of the effectiveness of these features and techniques would be helpful. We have revised the submission to include an additional Appendix E to quantitatively summarize how their effectiveness has been established from previous works when applied to Transformer variants. We also include some results applying one of the techniques, latent variable modelling similar to Chen et al. [1], on Graph2SMILES in Appendix E, as empirical evidence that its top-n accuracy can indeed be improved through established techniques at the expense of top-1 accuracy.
>
> We have had to defer exhaustive evaluations on whether Graph2SMILES can benefit from all of them because some of the codes either are not open-sourced yet (e.g. for EBM [2]), or only became available recently (e.g. for GTA [3] and Chemformer [4]). Nevertheless, we believe it is reasonable to hypothesize that Graph2SMILES can “potentially” benefit from at least some of them. Performance engineering may be the focus of a purely empirical follow-on report, but we feel that should not be a priority for ICLR.
>
> **Q2. “D-GAT does not outperform D-GCN in terms of the top-1 accuracy”.**
> We consider D-GAT and D-GCN simply to be design choices for the graph encoder rather than a major contribution of this work. We have updated Figure 1 as well as relevant sentences in Sections 1 and 2 to make this more explicit. As the reviewer correctly points out, the empirical comparison shows little difference between the performances of the D-GAT and the D-GCN variants. Therefore, we believe that it is sensical to leave both variants as possible design choices, both of which we have provided sample codes for and have included in the open-source release.
>
> **Q3. “Pairwise shortest path lengths … can be computationally prohibitive”.**
> We implement a variant of the Seidel’s algorithm [5] to compute the all-pairs-shortest-path from the adjacency matrix, which has a complexity of $O(n^w log \ n)$ where $n$ is the number of atoms, and $w < 2.38$ is the exponent in the complexity $O(n^w)$ of $n$ by $n$ matrix multiplication. This is therefore not the rate-determining step since the Transformer module has a complexity lower bounded by $O(n^2 d)$ where $d$ is the hidden dimension, and without the loss of generality, $O(n^3)$ since we set $d$ to $256$ which is typically much greater than $n$.
>
> **Q4. “The technical contribution is incremental”**
> Please see our response to all reviewers with the title “Reply to R1, R2, R3, R4: technical contribution is incremental”.
>
> [1] Chen et al. Path-augmented graph transformer network. http://arxiv.org/abs/1905.12712.
> [2] Sun et al. Towards understanding retrosynthesis by energy-based models. NeurIPS 2021.
> [3] Seo et al. GTA: Graph truncated attention for retrosynthesis. AAAI 2021.
> [4] Irwin et al. Chemformer: A pre-trained transformer for computational chemistry. DOI: 10.33774/chemrxiv-2021-v2pnn.
> [5] R. Seidel. On the All-Pairs-Shortest-Path Problem in Unweighted Undirected Graphs. Journal of Computer and System Sciences 1992.

---

> > ### Comment · Reviewer_3c52 · 2021-11-24
> > **Thanks for the response**
> >
> > I have read the authors' response and updated the rating accordingly. The supplementary experiment empirically demonstrates that existing techniques such as latent modeling indeed improve the retrosynthetic performance. Incorporating the discussion into the final revision would significantly improve the quality of this paper.

---

### Official Review · Reviewer_jeTQ · 2021-11-02

**Correctness:** 3
**Technical Novelty And Significance:** 3
**Empirical Novelty And Significance:** 3
**Recommendation:** 6
**Confidence:** 4

**Main Review:**

1.	From Eqn.(1) to Eqn.(5), you choose to use a complex gating mechanism to aggregate information. Is every component necessary? What if using a simple GCN or GAT?
2.	I think the model contains more parameters than conventional retrosynthesis models like conventional Transformer, GLN. Could you please show compare the number of parameters of different methods?
3.	The authors should provide some real cases to show how the method outperforms previous baselines, and why the method can obtain good results without templates.


Missing References:

1.	Dual-view Molecule Pre-training, https://arxiv.org/abs/2106.10234, the authors also work on retrosynthesis using Transformer and GNN models. A comparison is necessary.


**Summary Of The Paper:**

The authors proposed a new method for retrosynthesis, which does not require the mapping numbers and extracting templates from the literature.

Basically, the model consists of a graph-based encoder and a sequence based encoder. The encoder consists of local aggregation from neighbors and global attention using a new positional method. The decoder is a Transformer model with relative positional encoding.

The method achieved promising results on several retrosynthesis datasets.


**Summary Of The Review:**

The results in this paper are good, although the method itself is not quite novel.

---

> ### Author Response · Authors · 2021-11-21
> **Reply to R1**
>
> We thank the reviewer for the feedback and comments. We address each of them in turn.
>
> **Q1. “Complex gating mechanism… ​​What if using a simple GCN or GAT?”**
> The gating mechanism is not as complex as it looks; it merely replaces the RNN in message update with GRU [1], with $z_{ur}$ and $r_{uv}$ for the update and reset gates respectively, in a very similar form to the original GRU. This can be understood as a form of regularization, the effectiveness of which has been established in JT-VAE [2] and GraphRetro [3]. Since GraphRetro shows good performance for retrosynthesis, we simply follow their practice and adapt the same gating mechanism.
>
> We did not run the experiments for vanilla GCN or GAT because neither is designed to incorporate edge features (e.g., bond types). We did briefly experiment with an edge-enhanced version of GIN [4] as implemented in Pytorch Geometric, but chose not to include the results as the performance is worse than if the GNN is not used at all.
>
> **Q2. “Number of parameters”**
> Please refer to the following table for the parameter counts of certain models.  The numbers for Chemformers are taken from their paper, whereas those for GLN and RetroXpert are logged by us based on their codes, subject to minor deviations (e.g. due to non-deterministic vocab/semi-template size). There are too many Transformer variants to be included in the table, but the rows for GTA and Graph2SMILES (w/o D-GCN/D-GAT) can be thought of as representative of the Transformer families. Both share the same configurations (6 layers, 256 hidden size, 2048 ffn size), with slight difference due to different positional embeddings.
>
> | Model | #Params |
> |---|---|
> | GLN | 0.8M (GNN) |
> | RetroXpert | 2.3M (GNN) + 11.8M (Transformer) |
> | GTA | 17.6M (Transformer, est.) |
> | Graph2SMILES (w/o D-GCN/D-GAT) | 17.6M (Transformer) |
> | Graph2SMILES (D-GCN) | 0.4M (D-GCN) + 17.6M (Transformer) |
> | Graph2SMILES (D-GAT) | 0.7M (D-GAT) + 17.6M (Transformer) |
> | Chemformer (base) | 45M (Transformer) |
> | Chemformer (large) | 230M (Transformer) |
>
> For the methods summarized in the table, the GNN components have between 0.3M to 2.3M parameters, whereas the numbers for the Transformer components can range from 11.8M to 230M, or at least an order of magnitude larger. It then naturally follows that Graph2SMILES indeed has many more parameters than GLN which only uses GNN. For the similar reason, the inclusion of the D-GCN/D-GAT encoder in Graph2SMILES increases the parameter count only modestly over the Transformer-only Graph2SMILES (w/o D-GCN/D-GAT); an increase from 17.6M to 18.0M or 18.3M amounts to 2.2% or 4.0% respectively.
>
> **Q3. “Real case” comparison and reasons for “good results without templates”**
> We thank the reviewer for the suggestion to add real case studies. This will indeed be helpful for visualizing how Graph2SMILES has better encoding power than the Transformer baseline. We have revised the submission to include such cases for reaction outcome prediction in an additional Appendix F. Most notably, as compared to Molecular Transformer, Graph2SMILES seems to give better predictions for intramolecular ring formation, possibly by capturing the topological information explicitly on top of the local chemical environment (via the shortest graph distance feature).
>
> As for why our method, or template-free methods in general can sometimes work better than template-based methods, the question remains largely unresolved in the field. To the best of our knowledge, no prior work provides a rigorous answer other than empirical comparisons, possibly because template-free, end-to-end methods are less interpretable in general.
>
> **Q4. “Missing reference” and “comparison”**
> Thanks for pointing this out. Dual-view Molecule Pre-training [5] indeed evaluates on retrosynthesis. We have updated our Tables 2 and 3 to include relevant results, as well as Section 3.1 to include DMP as related work. To the best of our knowledge, in their retrosynthesis setup, the pretrained GNN branch is dropped completely in the DMP fusion setup, thereby making it closer to a SMILES-to-SMILES Transformer and dissimilar to our Graph2SMILES. This new comparison does not change our conclusion about the quantitative performance of the Graph2SMILES model relative to SOTA approaches.
>
> **Q5. “The method itself is not quite novel”**
> Please see our response to all reviewers with the title “Reply to R1, R2, R3, R4: technical contribution is incremental”.
>
> [1] Cho et al. Learning phrase representations using RNN encoder–decoder for statistical machine translation. EMNLP 2014.
> [2] Jin et al. Junction tree variational autoencoder for molecular graph generation. ICML 2018.
> [3] Somnath et al. Learning graph models for template-free retrosynthesis. ICML 2020 Workshop on GRL+.
> [4] Hu et al. Strategies for pre-training graph neural networks. ICLR 2020.
> [5] Zhu et al. Dual-view molecule pre-training. https://arxiv.org/abs/2106.10234.

---

> > ### Comment · Reviewer_jeTQ · 2021-11-25
> > **Reply**
> >
> > Thanks for your reply. The cases in Appendix F are good to me. The statistics of parameters is reasonable. However, the technical novelty is is mentioned by all reviewers, which prevents me to increase the score. I kept my score as 6.

---

### Author Response · Authors · 2021-11-21
**Reply to R1, R2, R3, R4: “technical contribution is incremental”**

We have updated the revised manuscripts with changes highlighted in blue.

Our work demonstrates for the first time that an end-to-end, permutation invariant graph-to-sequence model is able to outperform seq2seq models for molecular transformation tasks. This is a previously unverified hypothesis. Prior graph-to-sequence formulations in the past either cannot beat the simple Transformer baseline (as in GRAT [1]), or fail to achieve invariance to atom ordering by using the sequence representation in addition to the graph representation  (as in GET [2] and GTA [3]). It is true that elements of our graph encoder are drawn from existing works in different domains, but their combination and application are empirically novel.

The demonstration that a permutation invariant model can actually work has important implications for future research on these tasks. Many recent models [4]-[9] still make use of the vanilla Transformer architecture. These works rely on potentially costly input SMILES augmentation for competitive performance, but the decision on how many equivalent SMILES to generate for the input is non-trivial, as the performance has not been saturated even with 100 augmented SMILES [5]. Graph2SMILES removes this reliance on input augmentation altogether and yet outperforms the Transformer empirically. In this sense, our work provides evidence that Graph2SMILES is a viable drop-in replacement for the Transformer, in existing or future research efforts. We therefore believe that Graph2SMILES is empirically novel as a potential new backbone architecture over the vanilla Transformer for molecular transformation tasks.

[1] Yoo et al. Graph-aware transformer: Is attention all graphs need? https://arxiv.org/abs/2006.05213.
[2] Mao et al. Molecular graph enhanced transformer for retrosynthesis prediction. Neurocomputing 2021.
[3] Seo et al. GTA: Graph truncated attention for retrosynthesis. AAAI 2021.
[4] Schwaller et al. Molecular transformer: A model for uncertainty-calibrated chemical reaction prediction. ACS Central Science 2019.
[5] Tetko et al. State-of-the-art augmented NLP transformer models for direct and single-step retro synthesis. Nature Communications 2020.
[6] Irwin et al. Chemformer: A pre-trained transformer for computational chemistry. DOI: 10.33774/chemrxiv-2021-v2pnn.
[7] Yan et al. Retroxpert: Decompose retrosynthesis prediction like a chemist. NeurIPS 2020.
[8] Wang et al. Retroprime: A diverse, plausible and transformer-based method for single-step retrosynthesis predictions. Chem. Eng. Journal 2021.
[9] Sun et al. Towards understanding retrosynthesis by energy-based models. NeurIPS 2021.

---

### Decision · Program_Chairs · 2022-01-20

**Decision:**

Reject

**Comment:**

While the reviewers appreciated the method's ability to replace transformer models and SMILES data augmentation their main concerns were with (a) the experimental section, and (b) the technical innovation over prior work, which updated drafts of the paper did not fully resolve. Specifically for (a) this work performs very similarly to prior work: for reaction outcome prediction the proposed method improves top-1/3/5 for USPTO_STEREO_mixed but is outperformed by prior work for top-1/5/10 for USPTO_460k_mixed; for retrosynthesis the model is outperformed for USPTO_full and only outperforms prior work that does not use templates/atom-mapping/augmentation for top-1 on USPTO_50k. The authors argue that their method should be preferred because their method does not require templates, atom-mapping, and data augmentation. The reviewers agree that template-free and atom-mapping-free methods are more widely applicable. However, the benefits of being augmentation-free is not convincingly stated by the authors who only state that their approach is beneficial by "simplifying data preprocessing and potentially saving training time." The authors should have empirically verified these claim by reporting training time, because it is not obvious that their model which requires pairwise shortest path lengths is actually faster to train.
For (b) the reviewers believed that the paper lacked technical novelty given recent work (e.g., NERF). The authors should more clearly distinguish this work from past work (e.g., graphical depictions and finer past work categorization may help with this).
Given the similar performance to prior work, the lack of evidence to support training time claims, and the limited technical novelty, I believe this work should be rejected at this time. Once these things are clarified this paper will be improved.